# A scoping review of interventions on middle school students' attitudes towards science

**Noé Manuel García-Pérez[1,2]☯, Gonzalo Peñaloza [1]☯ ***

**1** Centro de Investigación y de Estudios Avanzados del IPN, Unidad Monterrey, Monterrey, México,
**2** Postdoctoral Researcher at Consejo Nacional de Humanidades, Ciencia y Tecnología (CONAHCYT),
México

☯ These authors contributed equally to this work.
* g.pjimenez@cinvestav.mx

scoping review of interventions on middle school
students' attitudes towards science. PLoS ONE
20(1): e0315757. https://doi.org/10.1371/journal.
pone.0315757

Biochemistry Leopoldo de Meis (IBqM) - Federal
University of Rio de Janeiro (UFRJ), BRAZIL

**Data Availability Statement:** All relevant data are
within the manuscript and its Supporting
Information files.

## Abstract

The aim of the study was to review the scientific literature on educational interventions to
promote positive attitudes towards science in middle school students. Due to the decline in
positive attitudes towards science observed in this critical age group of students and the
implementation of training programmes aimed at changing this situation, we sought to iden-
tify components of training proposals that have received attention from researchers. This
paper presents a scoping review of 37 papers published in English and Spanish over the
last 10 years. The review aims to describe the research outputs and analyse the effective-
ness and characteristics of program interventions, drawing from various databases. The
results show the need to examine the operational definition of the constructs used in the pro-
grammes, to focus attention on the influence of intrinsic variables on middle school students'
attitudes towards science, such as racial group or self-efficacy, and to conduct follow-up
evaluations to assess the permanence of attitudes. While acknowledging limitations related
to construct clarity or language restrictions in the search process the findings suggest that
middle school students' attitudes towards science remain an open area of research, as
there is no consensus on the characteristics of effective programmes.

## Introduction

While the cognitive dimension remains central in science education, as evidenced by the
implementation of curricula, selective teaching methods and reflections on students' learning
in terms of conceptual understanding, there is a growing emphasis in both research and educa-
tional systems on increasing the influence of affective factors in this process [1–4]. The special-
ised literature emphasises the importance of considering students' attitudes as a key
determinant of educational quality and effectiveness and advocates their cultivation in science
education [5–9].

There is a broad consensus in the research community on the central role of positive atti-
tudes towards science, which are considered to be a key determinant of variables such as aca-
demic achievement in the discipline [5, 10, 11], as well as long-term interest in science-related

**Funding:** This research was supported by the Consejo Nacional de Humanidades, Ciencias y Tecnologías (Conahcyt) of Mexico. These funds were received by Noé Manuel García Pérez.

**Competing interests:** The authors have declared that no competing interests exist.

careers [12, 13]. Nevertheless, fostering positive attitudes in individuals through formal education is recognised as a formidable challenge [14, 15].

Furthermore, it has been observed that this challenge becomes more pronounced during the secondary education phase, where not only is there no increase in positive attitudes towards science compared to earlier stages of education, but a decline is even observed as students get older or progress through school. This trend has been highlighted in several research efforts [16–19]. For example, in a comprehensive literature review, Christidou [20] concluded that the transition from primary to secondary school often coincides with a declining interest in science, with students increasingly perceiving it as an untenable career path. Similarly, a literature review that included an analysis of 56 publications investigating the effects of interventions using different teaching approaches on attitudes towards science and mathematics found a notable negative correlation between school year and attitudes towards science [21].

While it is true that this trend of changing attitudes in secondary education is a major challenge for the entire education system, there is also a large body of scientific literature claiming that attitudes can be positively influenced by various factors [8, 18, 22–26]. Among these factors, educational interventions seem to play a key role in their development [5, 19, 27, 28].

Building on this idea, Helvaci & Yilmaz [29] suggested that pedagogical practices developed by teachers have an impact on students' motivation and therefore promote the formation of positive attitudes towards science. In the same vein, Osborne et al. [5] highlight the specific characteristics of educational interventions as the main contributors to achieving positive attitudes towards science. Indeed, it has been found that negative attitudes towards science are mainly due to ineffective teaching methods, which are sometimes teacher-centred [28, 30], leading students to perceive science-related subjects as difficult or boring. With this in mind, it is important to identify the characteristics of educational interventions that achieve positive attitudes towards science among middle school students in order to increase the effectiveness of these interventions.

The purpose of the scoping review presented in this paper is to examine the scientific evidence available over the last 10 years on the characteristics of educational programmes that promote positive attitudes towards science in secondary school students.

These characteristics were analysed in terms of teaching approaches, duration, the person responsible for the intervention, the educational setting in which it was implemented, the monitoring of the development process, and the effects of the intervention on the attitude variable. As can be seen from the above statement, in contrast to previous literature reviews, which have mainly used quantitative methods to focus on and explain the results of educational interventions on attitudes towards science, the present study explored key features of educational interventions developed with the intention of understanding how they were designed and analysing gaps in the knowledge base.

In addition, the study focused on the construct of attitudes towards science as a whole, including data from disciplines such as physics, chemistry and biology, within the corresponding period of secondary education, a stage where a greater decline in this affective variable is observed and is considered fundamental for the development of scientific careers. In this context, the overarching question that guided this study was: What features are reported to be effective in educational programmes to promote positive science attitudes in middle school students?

## Factors influencing teaching science to middle school students

Educational intervention or instruction is understood as the process by which knowledge, skills and attitudes are deliberately and intentionally fostered, including the entire

instructional process, from planning and delivery to evaluation and feedback [31]. In science education in particular, the success of this process, that is, the ability of students to understand and engage in debates about science-related issues that shape our world, is crucial [12]. In this venue, it is important to consider a number of factors that support this educational action. Among these factors, instructional approach is particularly relevant in identifying effective science teaching practices. This is relevant because of the complexity of learning itself, as students require different types of explicit instructional support to understand and engage with the body of scientific knowledge [6].

There are many references in the literature to the effect that one or more teaching approaches can have on attitudes to science at different educational levels, among which those mediated by technological or virtual environments, based on inquiry strategies, out-of-classroom activities or cooperative learning, stand out [32–37]. For instance, a correlation between the achievement of positive attitudes towards physics and chemistry was found in a sample of Chinese students and cooperative learning strategies [38]. The impact of teaching approaches on students' attitudes towards science was also observed in the 2015 Programme for International Student Assessment (PISA). This analysis showed that attitudes towards science, including interest in scientific topics, enjoying science, and participating in science-related activities, showed better results with an inquiry-based approach compared to other approaches such as adaptive teaching or direct teacher instruction, despite the latter being associated with better academic performance [39].

Although teaching approaches are diverse and each has specific characteristics, they can be grouped according to their nature and intervention purposes [19, 21, 38, 40, 41]. To provide a more flexible and comprehensive framework for interpretation, teaching approaches found by Savelsbergh et al. [21] were adopted in this study and further supplemented by additional categories from Schroeder et al. [40]. This integration allows for a broader, more nuanced analysis of teaching strategies, accommodating the complexity and diversity inherent in science education. The categories are outlined in Table 1, with approaches 1 to 5 from Savelsbergh et al. [21] and approaches 6 to 11 from Schroeder et al. [40].

Table 1 shows the pedagogical approaches used in science education. It should be noted that some of the categories identified by the authors overlap in terms of the objectives of the pedagogical approach, even if they are not labelled as such. Some examples illustrating this situation are approaches related to inquiry-based learning, technology-mediated approaches and collaborative approaches. It was also found that the category "enhanced context strategies" identified by Schroeder et al. [40] could be included in the category "extracurricular" proposed by Savelsbergh et al. [21]. In all cases, the labels coined by the latter author were adopted.

In addition, it should be noted that the category of manipulative strategies, which refers to the teaching approach where the teacher provides opportunities for students to work with physical objects, also includes work with intangible elements, which mainly refers to declarative knowledge that can be represented tangibly, as in the case of a concept map.

## An overview of the available scientific evidence related to the object of study

A number of reviews have documented the effects of educational interventions on students' attitudes to science, indicating the interest of the academic community in this issue.

In this context, a literature review aimed at investigating the common factors that influence the attitudes of high school students towards chemistry documented that out of 36 studies found in the Eric and Google Scholar databases, 12 referred to teaching methods as a factor directly affecting student attitudes [18]. Of these studies, only one showed no positive effect on

**Table 1. Approaches to the teaching of science.**

| # | Teaching approach | Definition |
|---|---|---|
| 1 | Context-based teaching | One aim of all context-based curriculum is for students to experience the relevance and applicability of science content in society and in their personal lives (Gilbert, 2006). |
| 2 | Inquiry-based learning | In IBL, students engage in research activities to find answers to learning questions. The level of teacher guidance varies, but students take (partial) responsibility for formulating the research questions and methods, as well as for interpreting the results. |
| 3 | ICT-rich learning environments | ICT-rich teaching approaches include (individualised) computer-based instruction, games, feedback, interactive quizzes, computer-based laboratories, simulations and robotics. Suggested mechanisms include that students enjoy working with computers, students feel more confident to experiment and make mistakes, and/or students appreciate the (quick) feedback. |
| 4 | Collaborative learning | Collaborative and cooperative approaches to teaching, such as project-based work, discussion, jigsaw puzzles or peer feedback, tend to enhance social interaction and relationships between learners and often involve greater ownership of the content being learned. |
| 5 | Extracurricular | Extracurricular activities are not part of the standard curriculum or classroom environment, but are part of or closely linked to the school programme. Examples include field trips, mobile science laboratories, summer camps, guest lectures and visits to science centres. |
| 6 | Questioning strategies | Teachers vary the timing, positioning or cognitive level of questions (e.g. increasing waiting time, adding pauses at key points for students to respond, including more high cognitive level questions, stopping visual media at key points and asking questions, asking students comprehension questions at the beginning of a lesson or task). |
| 7 | Focusing strategies | Teachers make students aware of the purpose of the lesson or grab their attention (e.g. providing or reinforcing objectives in the middle or at the end of the lesson, using advance organisers). |
| 8 | Manipulation strategies | Teachers provide opportunities for students to work or practise with physical objects (e.g. developing skills using manipulatives or apparatus, drawing or constructing something). |
| 9 | Advanced materials strategies | Teachers modify teaching materials (e.g. rewriting or annotation of text materials, tape recording of instructions, simplification of laboratory equipment). |
| 10 | Assessment strategies | Teachers change the frequency, purpose, or cognitive level of testing/assessment (e.g. providing immediate or explanatory feedback, using diagnostic tests, formative tests, retesting, mastery testing). |
| 11 | Direct instruction | Teachers provide information orally or explicitly guide students through a series of tasks (e.g. learning by listening, designing experiments, using a microscope, making measurements). |

Table created by the authors.

attitudes as a result of the proposed intervention. Thus, the authors warn that the extent of attitude change may depend on the instruction provided during the learning process.

The results from the 2006 and 2009 PISA surveys were used to analyse a series of interventions developed in out-of-school contexts in six English-speaking countries (England, Wales, Northern Ireland, Ireland, Scotland and the United States) [42]. The aim was to investigate the participation of students under the age of 15 in these programmes, their academic achievement and their attitudes towards science, all in relation to the use of time. In general, findings revealead that students who stayed longer in out-of-school time (OST) programmes showed lower levels of academic achievement, in contrast to attitudes towards science, where a positive relationship was observed. While emphasizing the importance of time within a science content programme, the author concludes that there may be factors that play a more significant role in student outcomes, such as the quality and content of the programme or the type of intervention implemented.

In another systematic review of articles indexed in the Eric database published between 2000 and 2012, the objective was to investigate the relationship between kindergarten through

12$^{th}$ grade students (K-12) and science and technology. The authors of this review sought to identify what has been studied in terms of interest, motivation and attitudes towards these two disciplines. In the description of the literature reviewed, the study mentioned different subcategories highlighting educational interventions that showed positive results in the interest/motivation/attitudes (I/M/A) of the participants. From this, the researchers concluded that it is possible to increase students' I/M/A through different approaches or methods, as long as these are based on properly documented sources and consider other variables that come into play, such as the contextualisation of the content addressed or the pedagogical intention [19].

On the other hand, a meta-analysis was conducted to investigate how teaching approaches classified as innovative in science and mathematics affect the attitudes and academic performance of primary and secondary school students [21]. For the variable of attitudes, the analysis of 56 references obtained from the Web of Science, Scopus, ERIC and PsycINFO databases showed a significant positive effect of the interventions on general attitudes as well as on general interest and interest in science-related careers. However, no significant effects were found regarding the teaching approach, the duration of the intervention, the presence of teacher training, or the area in which each study was conducted.

In general, the data presented in the found literature reviews show the existence of positive results in attitudes towards science, academic achievement, interest or motivation of young students through the implementation of educational interventions. They also point out the relevance of the teaching approaches found as a factor influencing these results and provide a general overview of factors external to the educational intervention that influence the achievement of positive attitudes towards science. However, the nature of these studies did not allow for the precise identification of the characteristics of the interventions, so they did not identify other factors specific to the educational instructions that are relevant to their success, beyond the teaching approaches or their duration. The need to focus on the "quality" of the intervention, though not specifying which aspects are included in this term, has been highlighted in previous research [42].

In contrast, most of these literature reviews included other constructs in addition to attitude, such as interest and motivation towards science, or, due to the emergence of interdisciplinary learning approaches such as STEM and STEAM, included other fields in addition to science, such as technology and mathematics [29]. It has been suggested that studying the constructs together may lead to confusing interpretations of the outcomes of an intervention, making it relevant to address them independently [21]. Finally, the identified studies included students across a wide age range. This condition may hinder the identification of characteristics of effective interventions by educational level [43–47]. Therefore, it may also be appropriate to analyse only those educational interventions that are implemented at the appropriate age for secondary education.

## The purpose of this scoping review

Understanding the specificities of educational interventions in science teaching is relevant because it favours an in-depth analysis of the elements that promote good practice and enable the achievement, or lack thereof, of the stated objectives, in this case regarding the generation of positive attitudes. In the same context, the identification of these characteristics can help to compare and evaluate different teaching methods on the basis of their objectives, theoretical/conceptual and methodological foundations, rather than solely on the basis of their effects or results, in order to facilitate their replication in other contexts. Finally, the analysis of educational interventions makes it possible to identify factors that have been taken into account in these interventions and that may have a direct impact on secondary school students' attitudes towards science.

A scoping review of the characteristics of educational interventions aimed at achieving positive attitudes was conducted, focusing on documented classroom practices with secondary school students. It is expected that this literature review will help to identify constructs and aspects addressed in educational interventions as a step towards proposing models of good practice aimed at achieving positive attitudes in secondary school students.

In order to provide a more comprehensive overview of the selected literature, an attempt was made to address the following specific research questions (SRQs):

- How has the construct of attitudes towards science been approached in the literature in terms of focus, source references and scale dimensionality?

- What are the characteristics of the reported educational interventions in terms of teaching approach, duration, implementer, intervention context, effects on attitudes and follow-up of programme development?

- Within the educational interventions identified, what were the main problems and recommendations made by the researchers?

## Method

This literature review adopted the scoping review methodology proposed by Peters et al. [48]. This methodological design was chosen due to the need to scan the existing literature on the topic and expand knowledge beyond simply addressing the effectiveness of the intervention itself [49, 50]. Previous educational research using this approach has confirmed that this type of study can provide relevant information for research and educational practice in the context of changing attitudes towards science [51, 52].

The following search parameters were used to conduct a thorough search of the academic databases Eric, SciELO, Google Scholar, Dialnet, EbscoHost, Scopus and Web of Science for literature on educational interventions aimed at fostering attitudes towards science in high school students: studies focused on assessing the construct of attitudes towards science, involving the implementation of a training programme or educational intervention for students, focused on students between 12 and 16 years of age, conducted in the last 10 years and published in English or Spanish. As established, the sources of information for the scoping review could include any existing literature [53]. Therefore, the inclusion of literature was not limited exclusively to peer-reviewed primary research studies, but also included meta-analyses, systematic reviews and other scoping reviews. Although other types of literature, such as handbooks or books, were considered for inclusion in the corpus of documents, they were not found using the search criteria established.

The search was carried out during two months of 2023, specifically September and October, with the most recent exploration of literature conducted on October 21 of the same year. This process followed the methodology proposed by Peters et al., which consists of three main phases [48]. Each phase is described below, together with its implications for the development of the study.

1. Initial search in databases (DB). Starting from an initial search equation, a first exploratory exercise was carried out with the intention of finding concepts that would allow the refinement of the search equations. This phase was carried out in two databases, one in Spanish (SciELO) and another that would allow the retrieval of documents in English (ERIC). In the SciELO database, the initial search was performed with titles that contained one or more of the following terms: "actitudes hacia la ciencia" OR "interés hacia a ciencia" + "estudiantes de secundaria" OR "adolescents" + "intervención" OR "programa" OR "instrucción". In

contrast, the ERIC database was searched using English-language equivalents of these terms: "attitudes toward science" OR "interest toward science" + "middle school students" OR "adolescents" + "intervention" OR "evidence-based program" OR "instruction". The selection of these terms was aligned with the population, concept and context of the review, which are fundamental elements in a scoping review for establishing a priori inclusion criteria [48].

2. Subsequent DB search. This phase consisted of restructuring the initial search equations to search for primary documents for analysis in all the selected databases. Some keywords were included that refer to the breakdown of science into some of its disciplines (attitudes towards physics, attitudes towards chemistry, attitudes towards biology), as well as the keyword "secondary education" as a synonym for "middle school" in the English language. It was also emphasised that studies conducted with teachers or other populations that did not aim to develop and/or validate instruments to assess attitudes, or that only carried out an educational intervention, should not be included. In addition, after entering the search equations, some specifications were made in each database due to its characteristics. For example, in ERIC the temporality of the studies was specified, or in Scopus the search was restricted to social science studies. At the end of this phase, 133 documents were found.

3. Examination of the results. The third phase consisted of a manual selection based on the inclusion criteria specified in each of the DBs. This was done by reading the title, keywords and abstract of each document found. The full text was only consulted directly in those cases where the exposed elements could be clearly observed. After this step, 27 primary documents remained.

4. Searching reference lists. In order to ensure that material that was not included in the study and made an important contribution to the research was included, an additional step was taken to review the reference lists of review studies found in previous searches. These references also identified key journals that could contribute to the primary documents through manual review. A further 16 documents were located, but only 10 articles were included in the analysis corpus according to the inclusion criteria.

Where there was doubt about the inclusion or exclusion of retrieved texts, general agreement was reached within the research team on the basis of the inclusion criteria. For example, documents such as that by Houseal et al. reported the implementation of an intervention with students and teachers, so it was considered to include the study by reporting only the part related to the intervention with students [54]. In another case, that aimed to address attitudes towards other disciplines in addition to science was retained [21]. Furthermore, care was taken to ensure that only full texts were retained, excluding web pages, and that there were no duplicate articles.

The final outcome of the process was a total of 37 primary documents. It is worth noting that the elements of interest for the development of the study mentioned so far, such as objectives, research questions, inclusion criteria and methods of analysis, were specified and documented in advance in a research protocol (S1 Protocol). The literature search process is summarised in Fig 1.

## Screening process

Data extraction was carried out using an instrument or form that aimed to provide a logical and descriptive summary of the data, always ensuring alignment with the objectives and research questions raised in this scoping review (S1 Table). The data extraction instrument was based on the template suggested by the previously mentioned methodological guide [53]

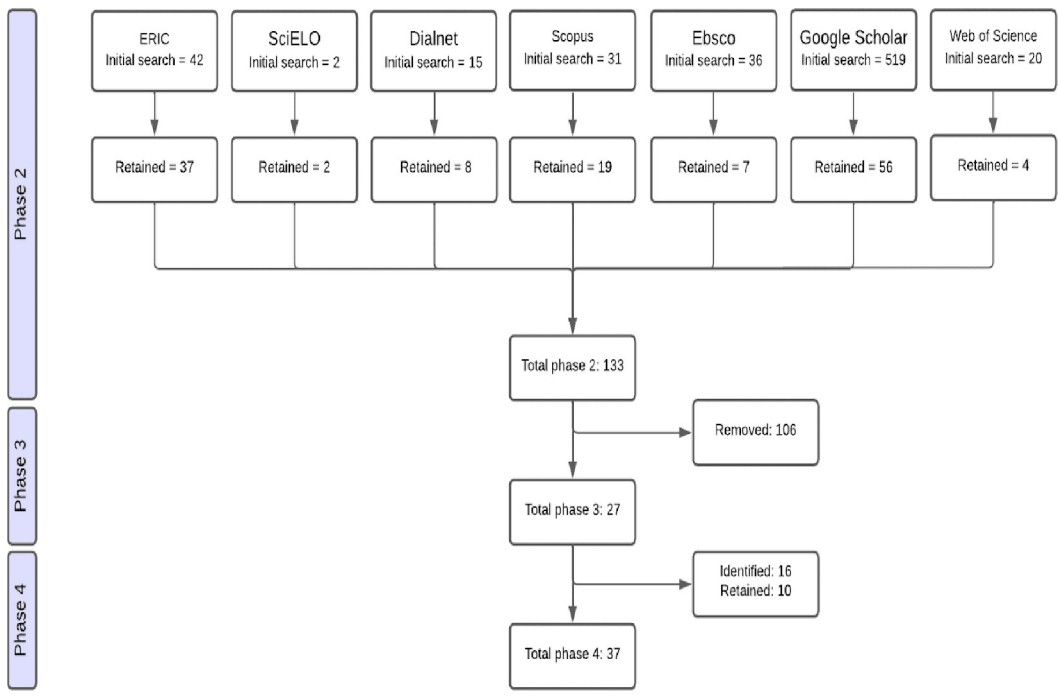

**Fig 1. Literature search process.**

and was validated by the research team through a pilot process that included reading some articles to capture their analysis in the instrument and to ensure a full understanding of each of its components.

At the end of this process, a refinement of the elements that would be part of the final instrument was achieved. In this sense, some data related to the section on procedures and suggestions for interventions to promote interest, included in the analysis network were reconsidered in relation to questions about the development of interventions [19]. This was done alongside the typology of teaching strategies drawn from previous studies [21, 40].

In addition, analysis columns related to the type of construct addressed were included. In summary, the final version of the instrument consisted of columns corresponding to information in four main categories: i) general information about the study, ii) about the construct, iii) about the target population, and iv) about the intervention. Where necessary, the information included in each category was coded to facilitate data analysis. The components of the final tool can be found in the attached tool in S1 Appendix.

## Results

Some characteristics of the studies in terms of demographic and methodological information (country of origin, year of publication, research design and methodology, number of participants and factors influencing students' attitudes towards science) and general components of the educational interventions found are presented before presenting the results in relation to the specific research questions posed. This information provides an overview of the information analysed in the current study.

The main characteristics of the educational interventions found are presented in Table 2. It should be noted that this table does not include the documents identified as literature reviews, as they do not allow the visualisation of elements of interest for this part of the research.

**Table 2. Approaches to interventions on middle school students' attitudes towards science.**

| Approach | Cite ID | Duration (weeks) | Person in charge | Inside/outside of regular classroom or a mix | Design assessment | Assessment of implementation process | Statement of problems/ recommendations | Effect |
|---|---|---|---|---|---|---|---|---|
| *1 Collaborative (CL)* | | | | | | | | |
| Cooperative learning | [55] | 12 | Teacher | Inside | No | Yes | No/ Yes | Yes (+) |
| Creative drama method | [56] | NM | Researcher | Inside | Yes | No | No/ Yes | Yes (+) |
| CL/ Virtual lab | [57] | 10 | Teacher | Inside | No | No | No/ Yes | Yes (+) |
| Innovative learning environment | [58] | 2 | Both | Inside | No | No | No/ No | Yes (+) |
| Science through sport | [30] | 4 | Teacher | outside | No | No | No/ Yes | Yes (+) |
| Cooperative learning for mastery | [59] | 4 | Teacher (s) | Inside | No | Yes (NS) | No/ No | Yes (+) |
| ABP, ADG, EPC | [60] | 8 | Researcher | NS | No | No | No/ Yes | Yes (-) |
| *2 Context-based teaching (CBT)* | | | | | | | | |
| Authentic learning | [61] | NM | Teacher | NS | Yes | No | No/ Yes | Yes (+) |
| Community service | [62] | 10 | Teacher | Mix | No | No | No/ No | Yes (+) |
| Citizen science | [1] | 28 | Teacher | Mix | No | No | No/ No | No |
| *3 Direct instruction (DI)* | | | | | | | | |
| Visually enriched/ Vocabulary/ Reading-Enhanced science instruction | [26] | 16 | Teacher | Mix | No | Yes (NS) | No/ Yes | Yes (+) |
| *4 Extracurricular (EC)* | | | | | | | | |
| OST learning environments | [63] | 18 | Teacher | Outside | No | No | No/ Yes | Yes (+) |
| | [64] | 7 | DNA | Outside | DNA | DNA | No/ Yes | No |
| Science program (SPICE) | [65] | 2 | Teacher | Outside | Yes | Yes | No/ Yes | Yes (+) |
| *5 ICT- rich learning environments (ICT)* | | | | | | | | |
| Simulation-Based instruction | [66] | 3 | Teacher | Inside | No | No | Yes/ Yes | No |
| ICT based narration activities | [3] | 6 | Teacher | NS | No | No | Yes/ No | Yes (+) |
| Application for mobile augmented reality | [67] | 8 | Researcher | Inside | No | No | Yes/ No | No |
| Digital storytelling | [68] | 6 | NS | Inside | No | No | Yes/ Yes | No |
| Virtual lab. | [69] | 12 | Researcher | Inside | No | No | Yes/ Yes | No |
| | [70] | 10 | Researcher | Inside | No | No | Yes/ Yes | No |
| | [71] | 4 | Teacher | Outside | No | No | No/ Yes | Yes (+) |
| CL/ Virtual lab | [57][a] | 10 | Teacher | Inside | No | No | No/ Yes | Yes (+) |
| Science teaching enriched ICT | [72] | 13 | Teacher | Inside | No | No | No/ No | Yes (+) |
| EBA/Experimental activities | [73] | 3 | Researcher | Mix | No | No | No/ Yes | Yes (+) |
| Virtual Lab/ Hands-on lab | [74] | 6 | Teacher | Outside | No | No | No/ Yes | Yes (+) |
| Visually enriched/ Vocabulary/Reading-Enhanced Science Instruction | [26] [a] | 16 | Teacher | Mix | No | Yes (NS) | No/ Yes | Yes (+) |
| *6 Inquiry-based learning* | | | | | | | | |
| Dialogic practical work | [75] | 12 | Teacher | Inside | No | Yes (NS) | Yes/ No | Yes (+) |
| Simulation-Based Instruction | [66] [a] | 3 | Teacher | Inside | No | No | Yes/ Yes | No |
| 5E´s learning cycle | [76] | 4 | Teacher | Inside | No | No | No/ No | Yes (+) |
| 5E´s learning cycle/ Gamification | [77] | NS | NS | Inside | No | Yes | No/ No | Yes (+) |
| Innovative learning environment | [58] [a] | 2 | Both | Inside | No | No | No/ No | Yes (+) |
| Experimentation activities | [78] | NS | Teacher | NS | No | No | No/ No | DNA |
| Argumentation-Based Science | [79] | 4 | Teacher | Inside | Yes | No | Yes/ Yes | Yes (+) |

*(Continued)*

**Table 2.** (Continued)

| Approach | Cite ID | Duration (weeks) | Person in charge | Inside/outside of regular classroom or a mix | Design assessment | Assessment of implementation process | Statement of problems/ recommendations | Effect |
|---|---|---|---|---|---|---|---|---|
| Student–Teacher–Scientist Partnerships (STSPs) | [54] | 5 | Teacher | Mix | No | Yes (NS) | No/ No | Yes (+) |
| *7 Manipulation strategies* | | | | | | | | |
| EBA/ Experimental activities | [73] [a] | 3 | Researcher | Mix | No | No | No/ Yes | Yes (+) |
| Visual Arts Education | [29] | 6 | Researcher | Inside | No | No | No/ Yes | Yes (+) |
| Virtual Lab/ Hands-on lab | [74] [a] | 6 | Teacher | Outside | No | No | No/ Yes | Yes (+) |
| Concept mapping | [59] | 4 | Teacher | Inside | No | Yes (NS) | No/ No | Yes (+) |
| *8 Not specified* | | | | | | | | |
| STEM supported science activities | [27] | NS | NS | Inside | No | No | No/ No | Yes (+) |
| Practical lab work | [80] | NS | Teacher | Outside | No | No | No/ No | Yes (+) |
| *9 Other* | | | | | | | | |
| 5E´s learning cycle/ Gamification | [77] [a] | NS | NS | Inside | No | Yes | No/ No | Yes (+) |

[a]Present in more than one of the categories

However, these documents are included in other analyses of the study. Following Table 2, the results are presented in different sections in order to answer the specific research questions posed. According to the adopted methodological proposals (Peters et al., 2020), the aim is to provide sufficient details on the characteristics of the concept, population and context of interest.

This scoping review focused on studies published within the last decade, as mentioned above. Fig 2 shows the results of the analysis based on the year of publication, where the continued interest since 2018 in addressing students' attitudes towards science through an educational programme is evident. Although no studies in this area were identified in the databases consulted in 2023, it is possible that publications occurred after the primary document search.

On the basis of the country in which the research was carried out or the data collected, the studies were categorised by continent of origin (Table 3). Due to the trend observed in the

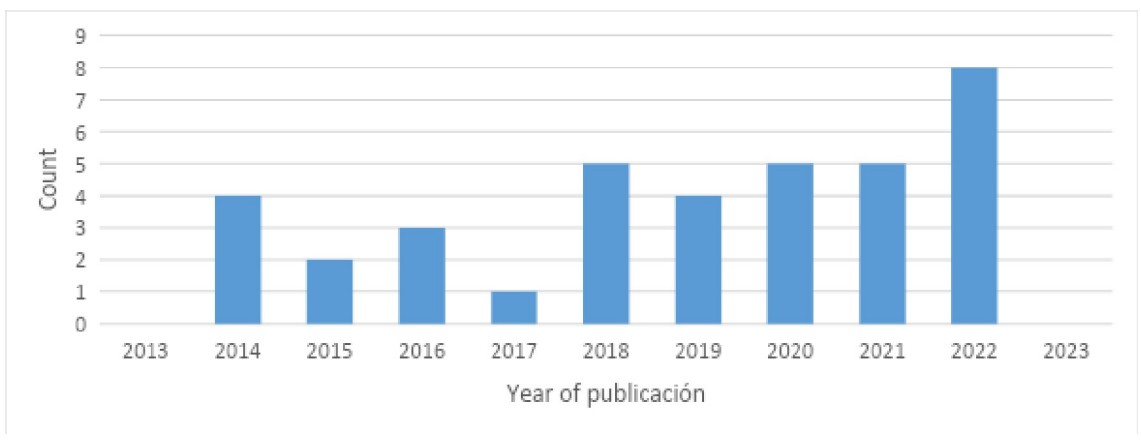

**Fig 2. Annual scientific production.** The figure displays the number of publications from 2013 to 2023. A notable increase in recent years, particularly in 2022, suggesting growing academic interest in this topic.

**Table 3. Number of publications per continent.**

| Continent | Number of publications | Proportion with regard to the body of documents (%) |
|---|---|---|
| Europe | 16 | 43.2 |
| North America | 8 | 21.6 |
| Asia | 5 | 13.5 |
| Africa | 5 | 13.5 |
| Central & South America | 3 | 8.1 |
| Australasia | 0 | 0 |

Americas, where the majority of publications were found in a minimum number of countries, it was decided to divide the continent into two parts: North and Central America on the one hand, and South America on the other. The research team felt that this decision would allow a better presentation of the results of the study in the first category of analysis.

Table 3 shows that the continent with the most contributions to this field of knowledge is Europe. Within this result, it is worth highlighting that the vast majority of studies were carried out in Turkey (14 studies). After Turkey, other countries that contributed significantly to this topic were, in decreasing order, the United States (6), Rwanda (3), Canada (2) and Colombia (2). Countries such as Ethiopia, Bhutan, Thailand, Oman, Nigeria, Qatar, Chile, Portugal, Lebanon and the Netherlands contributed only one document each to the corpus (Fig 3).

In terms of research design, which is defined as the plan or proposal for guiding research that involves the intersection of philosophical underpinnings, research strategies, and specific methods [81], it was found, as shown in Fig 4, that studies approached from a quantitative perspective predominated, with 23 of the located documents falling within this subcategory. Eight studies located within the mixed methods subcategory and two within the qualitative subcategory completed the sample. The analysis shown in Fig 4 does not include documents that aim

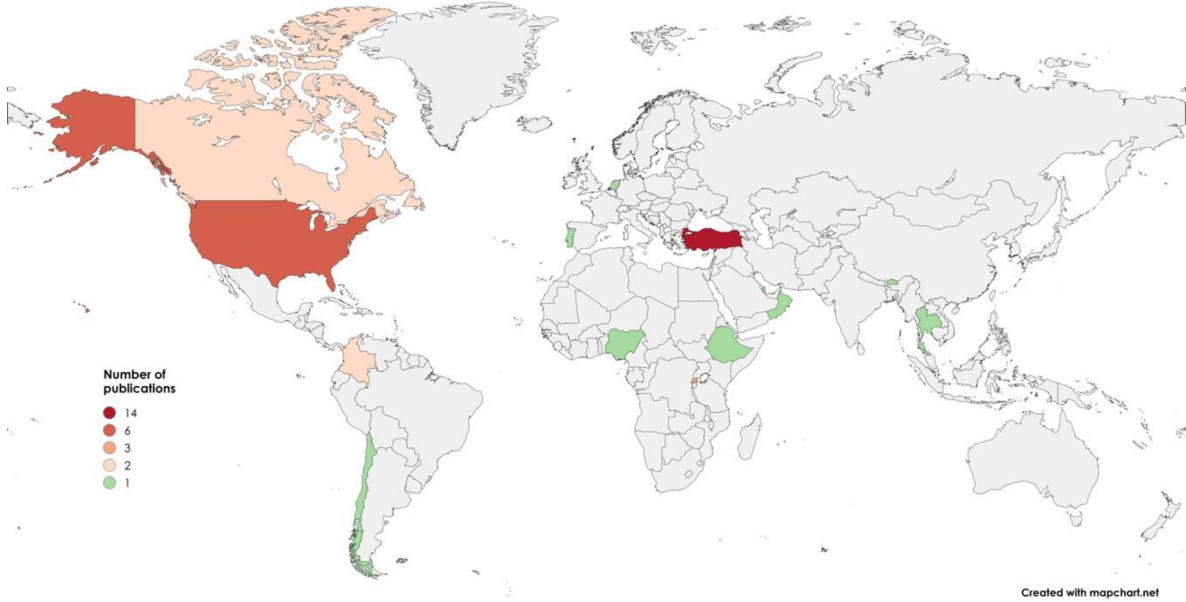

**Fig 3. Distribution of the publication per country.** Countries are color-coded according to the number of publications, with darker shades representing a higher number of publications. The scale at the bottom ranges from 1 to 14 publications, with the highest concentration observed in Turkey. Following Turkey, the USA and Rwanda appear in lighter shades. We acknowledge the use of MapChart (https://www.mapchart.net) for creating Fig 3 in this article under a Creative Commons Attribution-ShareAlike 4.0 International License.

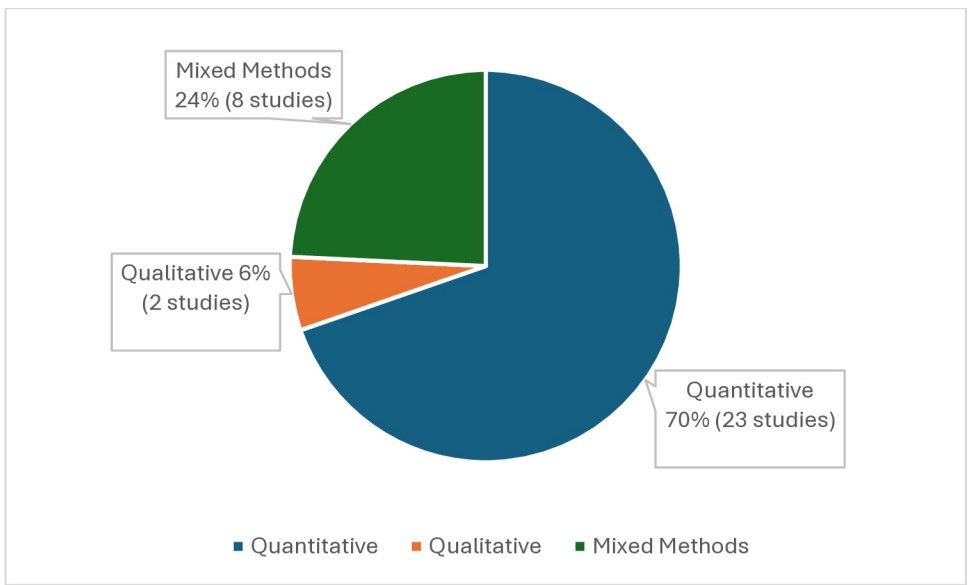

**Fig 4. Research designs.** The pie charts presents the distribution of research designs reported by the main authors of the studies. This distribution highlights the preference for quantitative approaches in the field.

to conduct a literature review. It only includes documents that describe the implementation of an educational intervention and/or report on its results. Therefore, the total number of studies included in the analysis of both designs and research methods is 33.

From the different research designs presented, the research strategies or methodologies found are reported, understood as "types of quantitative, qualitative and mixed methods research designs that provide specific guidance in a research design" [81]. The findings are summarised in Table 4.

Studies using a quasi-experimental research methodology account for the vast majority of research within a quantitative design and total number of documents. They are followed by studies that used a pure experimental design. The document that reported the implementation of a survey [64] remains in this literature review as it addresses the characteristics of interest within an educational intervention.

**Table 4. Research methods.**

| Applied method | Number of cases | % related to research design | % related to total |
|---|---|---|---|
| *Quantitative research design* | | | |
| Quasi-experiment | 18 | 78.2 | 54.5 |
| True experiment | 4 | 17.3 | 12.1 |
| Survey | 1 | 4.3 | 3 |
| *Qualitative research design* | | | |
| Case study | 2 | 100.0 | 6.1 |
| *Mixed methods research design* | | | |
| Convergent Parallel Design | 3 | 37.5 | 9.1 |
| Exploratory Secuencial Design | 1 | 12.5 | 3 |
| Embedded design | 1 | 12.5 | 3 |
| Not specified | 3 | 37.5 | 9.1 |

In terms of mixed methods research design, the findings were categorised based on a proposed framework [82]. Three different research designs were identified. However, three documents [26, 55, 65] did not provide a categorisation in terms of the research design conducted.

On the other hand, there was a great diversity both in the age and school years of the students participating in the interventions found, as well as in the number of students involved in each of them. It is worth noting that the education systems from which the data in this study are derived have different organisation in terms of the grades considered for secondary education, so that in some cases they are not equivalent to each other. It is therefore difficult to synthesise and present the data found in relation to the school year.

There is also a significant difference in the number of participants in each intervention. In order to establish a point of analysis, a distinction can be made based on the research design. It was found that the minimum number of students involved in an intervention was 15 in a study that used a mixed research design [30]. Similarly, the study with the largest number of students involved reported a total of 442 participants [1]. Considering only the studies with a quantitative research design (quasi-experimental and true experimental methods), the average number of participants in the 22 studies included in this analysis is 100. However, it is important to accentuate that some of the documents analysed reported the inclusion of more than one experimental group due to the involvement of more than one independent variable [57, 59, 60, 74].

Finally, with the aim of exploring the characteristics of secondary school students, an attempt was made to identify factors that have been underscored by researchers as elements that may influence attitudes towards science within an educational intervention, without the need to intervene in them. In the case of the literature review studies included in this section, the general conclusions highlighted by the researchers in each of the identified variables are presented. Table 5 shows the results obtained based on the construct addressed by each of the researchers in their studies.

**Table 5. Factors influencing the attitudes of students in the context of an educational intervention.**

| Factor (s) | Cite ID | Construct | Effect |
|---|---|---|---|
| *Student-related variables* | | | |
| 1 Sex/Gender | [75] | Attitudes towards Physics | No |
| | *[18] | Attitudes towards Chemistry | No |
| | [57] | Attitudes towards Biology | No |
| | [64] | Attitudes towards STEAM | No |
| | [59] | Attitudes towards Biology | No |
| | *[19] | Attitudes towards science | No |
| 2 Scholar year | *[18] | Attitudes towards Chemistry | Yes (-) |
| | *[19] | Attitudes towards science | Yes (-) |
| | *[21] | Attitudes towards science | Yes (-) |
| 3 Socioeconomic level | [67] | Attitudes towards science | No |
| | [64] | Attitudes towards STEM | No |
| 4 Racial group/Ethnicity | [64] | Attitudes towards STEM | No |
| 5 Academic performance | [75] | Attitudes towards Physics | Yes (+) |
| | *[19] | Attitudes towards science | Yes (+) |
| 6 Self-efficacy | *[19] | Attitudes towards science | Yes (+) |

Yes (+), Factor with positive effect; Yes (-), Factor with negative effect; No, Factor with no effect.
*Literature review.

Six factors were identified which were measured in the attitudes of secondary school students and which were of interest in 8 of the 37 documents included in this study (21.6%). The variables that received the most attention were those related to sex / gender (6 studies) and those related to school year (3 studies).

### Research question 1: How has the construct of attitudes towards science been treated in the literature in terms of approach, source references and scale dimensionality?

It has been noted that defining attitudes towards science encompasses a multitude of concepts that refer to activities carried out inside and outside the classroom, as well as different components that can be included in unidimensional or multidimensional constructs [8]. As a result, this phenomenon becomes a construct that encompasses a great deal of complexity. For this reason, an attempt was made to identify the general characteristics of the construct in the primary documents obtained, as well as the main theoretical references that support each of its definitions.

In a preliminary analysis of the construct of attitudes towards science, it was found that only seven of the total identified studies referred to a conceptual definition of the construct of interest (18.9%). Table 6 summarises the definitions provided by the authors of the primary interventions.

Table 6 shows the agreement between authors in identifying the construct under study as a phenomenon composed of different components, specifically affective, cognitive and instrumental. However, only in one case is there a reference to the dimension or dimensions that make up the construct [75]. In the vast majority of these cases, a general definition is used, even though the studies focus on specific disciplines. The remaining 30 studies (78.4%) simply emphasise the need to study this variable without directly defining what is meant by attitudes towards science.

As can be seen from the analyses presented, the review of the literature made it possible to identify the differences that existed in the corpus of documents with regard to the construct under study, which allowed a distinction to be made between each of them. As a result, the nature of the construct was investigated, specifically the object of interest toward which the attitudes are directed. Based on consensus in the scientific literature [5, 8], we identified articles addressing attitudes toward the following objects:

- A unitary concept of science (attitudes toward science in general);

- An integrated concept of scholar science course (attitudes toward a science course);

- A specific discipline such as biology, physics or chemistry (attitudes toward a science discipline);

- A particular topic within a discipline or scholar course, such as "evolution" or "cell functions" (attitudes toward a particular topic);

- Other.

Fig 5 shows the prevalence of studies that addressed the construct of attitudes towards science in a general way. It can also be seen that a significant proportion of these studies attempted to investigate attitudes towards some of the scientific disciplines covered by the study. In this category, the same number of studies was recorded for each of the disciplines covered (physics, chemistry and biology), with a total of 4 for each. One study investigated attitudes towards physics and chemistry [1], according to the authors, due to the characteristics of

**Table 6. Definition of the construct under study.**

| Cite ID | Definition of the construct | Adapted from |
|---|---|---|
| [1] | The feelings, beliefs and values held about an object, which in relation to science may include enthusiasm for science, perceptions of school science, and the contributions of science to society or of scientists themselves. | Osborne, Simon, & Collins, 2003 |
| [75] | The feelings and values that students have towards physics in terms of the dimensions: enthusiasm for physics, learning physics, practical work, physics teacher and future profession, expressed in terms of like or dislike. | Kind et al., 2007; Kaur and Zhao, 2017 |
| [59] | The predisposition of a person to think, feel, or prefer something in response to his or her beliefs about the object, which may be positive or negative. Attitudes are cognitive, emotional and action tendencies towards a particular behaviour. | Jain, 2014; Oghenevwede, 2019; Mazana et al., 2018; Hacieminoglu, 2016. |
| [60] | Attitudes are expressed through three components: the cognitive, which includes beliefs, thoughts, concrete evaluations, facts and opinions about a person, situation or object; the affective-evaluative, which refers to the processes that support or contradict our beliefs (feelings, preferences, moods, emotions); and the instrumental or behavioural, which refers to the assessment we make of the possibility of using the knowledge we have learned in the future. | Palomino, 2013; Gargallo et al., 2007. |
| [18] | A combination of individual values, feelings and beliefs towards science. | Hacieminoglu, 2016; Montes, Ferreira, & Rodríguez, 2018; Morabe, 2004; Salta & Tzougraki, 2004. |
| [21] | A psychological tendency that is expressed by the evaluation of a particular entity with a degree of favour or disfavour. | Eagly & Chaiken, 1993 |
| [26] | The feelings, beliefs and values held about an object, which may be the enterprise of science, school science, the impact of science on society, or scientists themselves. | Osborne, Simon, & Collins, 2003 |

Original creation.

the science curriculum in the country where the research was conducted, hence there is no alignment between the count of the subcategory Attitudes towards a science discipline and the individual count of each discipline.

Regarding the results in the "other" subcategory, studies were found that addressed attitudes towards the STEM (Science, Technology, Engineering, and Math) proposal, including Arts (STEAM), or studies that, based on the authors' definition, addressed attitudes towards science learning.

Within the general structure of the construct, the aim was to determine whether the attitude variable consisted of one or more dimensions. In this regard, 21 studies (56.7%) used a multi-dimensional construct, 6 (16.2%) were based on a unidimensional construct, 8 studies (21.6%) did not provide details on the specific characteristics of the concept under investigation, and 2 other studies (5.4%) were not applicable for this analysis. For studies based on a multidimensional construct, it was examined whether the results were presented by dimension. It was found that 13 studies presented this type of data, representing 35% of the total sample.

For this study, it was of interest to identify the references of the attitude constructs mentioned in the documents. On this basis, three categories were established: 1) when the

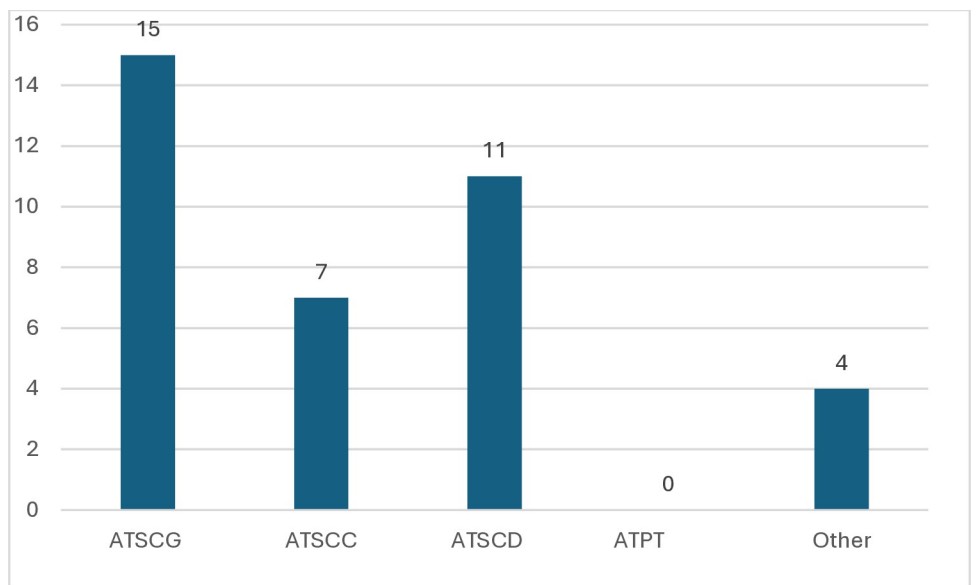

**Fig 5. Focus on the construct or concept of interest.** The main constructs addressed in the studies include: attitudes toward science in general (ATSCG), attitudes toward a science course (ATSCC), attitudes toward a science discipline (ATSCD), attitudes toward a particular topic (ATPT), and constructs that do not fall into any of the mentioned categories (Other).

theoretical proposal/conceptual definition was taken from only one reference; 2) when the theoretical proposal/conceptual definition was based on an adaptation from different authors; and 3) studies that did not specify the origin of the term or that did not apply to such an analysis.

In Fig 6, it can be seen that the documents reporting a study that focused on the construct of attitudes towards science tended to adopt a conceptual definition from a single author, as 10 out of 15 studies (66.6%) that addressed this construct did so. For studies that investigated attitudes towards a science course, it was found that all of them used only one reference (seven documents). As far as the documents presenting research on attitudes towards a specific discipline are concerned, there is a diversification in the origin of the references to the phenomena of interest, since five studies (45.4%) were based on an adaptation by more than one author. This subcategory was the only one that presented this phenomenon. Finally, the studies dealing with attitudes towards other types of constructs showed a tendency to adopt the position of a single reference.

On the basis of the references found, it can be noted the coincidence of two theoretical-conceptual proposals in the documents that were part of the analysis: 1) that of Benli´s proposal, originally from 2010 and cited in his later work [27] on the construct of attitudes towards science, which is also referenced in another study [3]; and 2) that of Prokop et al.´s proposal [83], on attitudes towards biology, which is cited in three documents [57, 59, 76].

On the other hand, an examination was made of the information gathering instruments used to estimate the variable related to students' attitudes. It was found that all studies, with the exception of those that aimed to conduct a literature review, used a questionnaire or an inventory with a Likert scale, usually of 5 points, as an information gathering instrument. Some studies complemented the information collection process with other instruments, such as two studies that reported the use of semi-structured interviews [26, 30] and one that indicated the use of an unstructured interview [62]. In addition, the use of an observation format

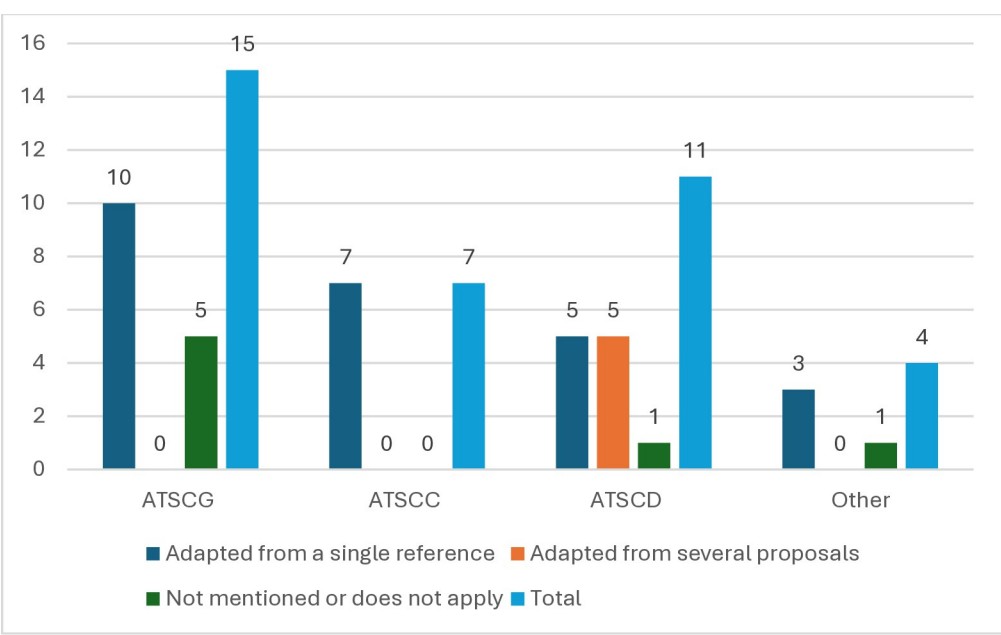

**Fig 6. The origin of the theoretical and conceptual foundations of the construct of interest.** Source of theoretical-conceptual support for the attitude constructs of interest. This figure illustrates the origins of the references used for each attitude construct mentioned in the reviewed studies. The constructs include attitudes toward science in general (ATSCG), attitudes toward a science course (ATSCC), attitudes toward a science discipline (ATSCD), and constructs that do not fall into any of the mentioned categories (Other). Total references per construct are also displayed.

as an information-gathering tool was documented, but the aim was not to assess students' attitudes towards science but rather to examine the extent to which the designed intervention was characterised by the proposed methodological strategy.

### Research question 2: What are the characteristics of the reported educational interventions in terms of approach, duration, provider, intervention context, impact on attitudes and monitoring of programme development?

The variety of teaching approaches proposed to develop students' attitudes towards science was an element of interest for the present study. An attempt has therefore been made to present the data in this section according to each of the approaches identified in the body of knowledge. These approaches reflect the nature of the intervention, following the categories previously cited [21, 41]. Table 2 shows the pedagogical approaches used in each educational intervention. Note that in some cases these approaches appear in more than one category, as the primary researchers used more than one teaching method, either in one or more experimental groups. It should be clarified that not all science teaching approaches found in Table 1 were included.

The analysis of the corpus of documents revealed seven different teaching approaches identified in the literature, based on the proposed categorisation. In addition, two other categories were found: the "not specified" category, in which two studies were found [27, 80] that, although they propose an intervention from which they introduce the type of content, objectives and teaching approach used, do not describe the specific characteristics of each of these elements; and the other category, in which one study was found [77] that does not fit into any of the proposed categories. In their intervention, these authors use two approaches within their intervention, so they can be found in two different categories.

Table 2 shows that the pedagogical approach most frequently used by the authors is that which refers to learning environments mediated by information and communication technologies (10 studies, 30.3%). However, it should be noted that this approach is the one that shows the highest number of interventions with no effect on students' attitudes towards science. This approach is followed by inquiry-based learning and collaborative approaches. On the other hand, the category with the fewest proposals is related to direct instruction.

The duration of the interventions varied considerably. Using weeks as a reference point, it was found that educational instructions ranged from two weeks [58, 65] to 28 weeks [1]. However, the average duration of the educational interventions included in this study is eight weeks.

Overall, there was a prevalence of interventions implemented by subject or group teachers (21 studies, 63.6%). Seven interventions (21.2%) were developed by the researchers in charge of the intervention, and only in one case (3%) was the intervention developed jointly by the group teacher and the researcher. In four other cases, the person in charge of the educational instruction was not specified. In terms of setting, the majority of interventions took place in the classroom (18 studies, 54.5%), while seven interventions (21.2%) took place outside the classroom, in places such as sports fields, computer rooms or laboratories, and a further five studies used a mixture of indoor and outdoor settings (15.1%).

The remaining three studies (9%) did not provide this information. Regarding the monitoring of the development of the educational intervention, which included an evaluation of the programme design and whether there was an evaluation of the implementation process based on the pedagogical approach used or the objectives of the study, it was found that four studies (12.1%) covered the first indicator, mainly using the judgement of experts in the field of science education or the pedagogical approach used. On the other hand, eight documents (24.2%) reported the existence of monitoring of development, carried out through observation formats to assess the extent to which lessons were characterised by the components of the same pedagogical approaches. However, only two studies detailed this information [55, 65].

Finally, 25 of the studies reviewed (75.75%) reported a positive effect on attitudes towards science as a result of the interventions implemented. This result is consistent with the data presented in previous literature reviews [18, 19, 21]. Only one study reported a negative effect on students' attitudes as a result of the intervention carried out [60].

## Research question 3: What were the main problems and recommendations identified by the researchers in the educational interventions that were found?

A final point of interest was the identification of problems and recommendations made by researchers in each of the studies, as this feedback is valuable for strengthening science education. Identifying problems and recommendations in educational instructions can ultimately become a crucial step in promoting the idea of solutions or innovative ideas tailored to the characteristics, needs and capacities of each educational context.

This section only collects elements related to the educational intervention process, so recommendations on research methodologies are not taken into account. Table 2 in the category of problems/recommendations reflects this situation. Based on the data found in the corpus of primary documents, the following are the common themes that give rise to both problems and recommendations put forward by researchers.

**Duration of the intervention.** This topic received the most attention from researchers, with eight studies addressing this element either in the problems identified or as part of their recommendations. Authors emphasised the need to extend the time allocated to the

educational intervention, either because of the difficulty of influencing attitudes towards science [67, 68] simply because they considered a few weeks insufficient for its development [69, 79].

**Teacher training.** Another important theme within the analysis was the training that teachers receive to implement and evaluate the proposed education. In total, four studies highlighted this element as part of their problems/recommendations. The authors suggest that teachers need support to prepare and implement the selected teaching approach in their classes in order to foster positive attitudes towards science [61, 79] or to achieve effective implementation of the proposal [66].

**Student training.** Some authors identified student training as an area of opportunity to achieve better outcomes in the intervention. It has been noted that the participating students were not familiar with the programme, the teaching approach or the technology used [67, 69, 79] which could have affected the development of positive attitudes towards science. Three studies contributed to this topic of interest.

**Evaluation of proposed activities.** Two studies accentuated the need to evaluate the set of activities to ensure their relevance to the intervention objectives. For example, one study [61] mentioned the need to review the activities, while another [29] added the importance of having an expert in the field during the research, although no details were given.

**Other issues.** In addition to the themes discussed, other issues were mentioned by the researchers. The language used by the instructor [70], the inclusion of more than one topic in the experimental activity [74], the inclusion of formal and informal learning contexts in the same programme [63], and the cost of the intervention programme [3] were elements identified as points of reflection for the success of the teaching programmes.

## Discussion

The review of 37 primary documents published in the last decade has allowed a deeper understanding of the key characteristics that define educational interventions, emphasizing the socio-demographic variables of importance, the theoretical-conceptual underpinnings and the facets of the proposed instructional designs. In addition, an attempt has been made to identify the main challenges and recommendations identified by authors in the course of intervention development, culminating in the identification of elements that guide best practice in the promotion of positive attitudes towards science.

The primary finding of this study highlights a prevalent trend in intervention programmes aimed at cultivating positive attitudes towards science: a lack of examination or delineation of specific components inherent in the instructional design, with a predominant emphasis on the outcomes achieved. This observation aligns with findings from a literature review indicating that the majority of the studies reviewed reported outcomes related to students' attitudes, motivation and interest mainly through questionnaires or interviews, without delving into an evaluation of the educational interventions in terms of their constitutive elements [19]. The most common category in this analysis, referred to as 'portraits of students' interest, motivation and attitudes', focused on measuring the impact of interventions on these variables, relegating the analysis of the characteristics of educational interventions to a secondary role for primary researchers. While this review covered a broader scope in terms of the constructs examined, it serves as an indicative reflection of the prevailing trend observed to date regarding intervention strategies aimed at shaping students' attitudes. The latter construct, along with interest and motivation, is recognised as an integral part of the affective dimension of learning [1, 21, 67].

It's worth noting that more than half of the studies examined were carried out exclusively with students from just two countries: Turkey and the United States, together accounting for

51.3%. In addition, only a small number of studies were conducted in Latin America (3). These findings raise important questions about whether the concentration of research in specific countries has influenced the results in terms of attitudes towards science and, ultimately, the level of scientific literacy of the participants—a goal expected by the international education community and enshrined in various references that describe the competencies students should have in science by the age of 15 [12].

Furthermore, these findings prompt reflections on the establishment of a well-supported field of science education research (SER) on these countries, a trend previously identified in other literature reviews [19, 51]. The prominence of these two nations in the SER landscape has been consistently stressed, with both being ranked among the top 10 most productive countries in the field [84, 85]. Several factors may explain this trend, including the emergence of initiatives aimed at improving science education, the early development of science educations as an academic discipline, the presence of graduate programs specializing in this area, or financial support for researchers [85]. In the case of Turkey specifically, studies on attitudes and perceptions has been reported as one of the top three most frequently investigated topic by science education researchers. This area of research may also be influenced by the growing interest among Turkish scholars in publishing in international journals, driven by national promotion policies [86].

Most importantly, the results underscores the necessity of capturing the diversity of educational contexts in scientific literature, especially regarding the specific characteristics of science education interventions that other regions. We also acknowledge that this absence of information may be due to research from other areas being reported using different indicators. This issue will be revisited in the next section.

Quantitative research designs, including quasi-experimental and true-experimental methods, were also found to be predominant in the exploration of this research area. The prevalence of these approaches has several advantages, including the use of robust statistical techniques and the early detection of potential threats to the internal validity of experiments. However, unlike previous literature reviews that have predominantly examined quantitative research findings, the data presented in this study provide insights that are crucial for understanding the intricacies of interventions aimed at cultivating attitudes towards science, an area of research previously highlighted as lacking [5].

In terms of student demographics, an average enrolment of 78 students was found across the interventions, with instances of students being allocated to multiple experimental groups [57, 59, 60, 73, 74]. There were no discernible correlations between attitudes towards science and variables such as gender, socioeconomic status or ethnic background. However, a notable negative correlation was observed with school year, as discussed above. Conversely, positive effects were found for variables related to academic achievement and self-efficacy. While these findings are valuable in identifying gaps in the factors influencing students' attitudes, it is advisable to treat such findings with caution. Firstly, the data includes a diverse age range within the student population [19], and secondly, some reports come from studies that go beyond attitudes towards science alone to include a wider range of variables or subjects [64]. Both of these factors may significantly influence the results obtained.

Among the studies analyzed, few specifically addressed the barriers faced by minorities in engaging with science, despite existing research showing disparities in science engagement among certain social groups, such as women, Black, and Latinx individuals [87]. While some studies did not report significant differences, this could be attributed to methodological limitations, such as the absence of intersectional and longitudinal approaches or insufficient sample sizes to analyze diverse identities [88]. These limitations may lead to an underestimation of the obstacles faced by disadvantaged groups and provide an incomplete understanding of the

external factors that influence attitudes toward science [89]. In fact, some studies suggest that structural discrimination may contribute to negative attitudes toward education [87] and limit access to resources that could enhance scientific engagement. Additionally, the intersection of race, gender, and socioeconomic status often exacerbates these challenges, perpetuating a cycle of underrepresentation in science and technology fields [90].

Adopting an intersectionality approach that includes a broader range of factors in the analysis could be more effective in identifying how ethnicity, class, gender, and other social categories influence the relationship between certain communities and science [91]. For example, some studies have demonstrated differences in science engagement between Latinx boys and girls, as well as between youth in urban and suburban areas [90]. This suggests the need for an approach that incorporates multiple intersecting factors. Furthermore, studies conducted in diverse contexts and populations are essential for understanding the ways in which underrepresented students engage with science [91]. In summary, there is a pressing need for research that gathers robust validity evidence on the experiences of underrepresented groups in science education.

The study has revealed a divergence in the conceptualisation of attitudes, depending on the specific object towards which these attitudes are directed. Apart from authors who emphasise the construct within disciplines that include science (such as chemistry, physics and biology), notable differences emerge when attitudes towards science are assessed in a broader context. This includes students' feelings and values about scientific endeavour, which are not limited to a specific academic subject, in contrast to attitudes towards science courses, which are more oriented towards activities or practical engagements within the school environment. This finding resonates with previous findings that underlined the potential disparity between attitudes towards science in a general sense and those experienced within the confines of formal education, highlighting the need to recognise this distinction [8]. Nevertheless, the current research calls attention to instances where the delineation of this boundary proves to be complicated, as certain studies examining attitudes towards science include factors related to enjoyment of science education [54, 66, 78].

The above observations may suggest, as mentioned earlier, a lack of consensus on the conceptualisation of the construct, pointing up a significant opportunity to adopt precise theoretical definitions in the study of attitudes. This ambiguity is reflected in certain findings. For example, some studies use the constructs of attitudes towards science and attitudes towards science education interchangeably, leading to potential confusion about the phenomenon under investigation. In addition, a notable feature of these studies is the wide range of theoretical frameworks employed by researchers; despite the relatively modest size of the corpus of primary documents in this review, congruence in theoretical-conceptual propositions was found in only two specific instances. A similar pattern emerges with regard to the tools used to collect data, each of which incorporates different factors or dimensions. Finally, the limited percentage of documents that provide a concrete definition of the attitudinal construct suggests that this variable may play a subordinate role compared to the priority given to other variables in these studies, such as academic achievement.

The top three teaching approaches identified, in descending order of prevalence, were: immersive learning environments rich in information technologies, research and collaborative elements. However, while it is evident that researchers can improve attitudes towards science through the use of electronic devices, software or computer-mediated interventions—due to their ability to facilitate the creation of mental models and enable direct observation of phenomena that are otherwise difficult to access in real, controlled and authentic contexts, alongside the use of various online scaffolding tools [26, 74] the optimal effectiveness of this teaching strategy has yet to be conclusively demonstrated.

In this context, various aspects integral to the design and implementation of each proposal were examined, including the duration of the interventions. Although this factor has been recognised as crucial for promoting positive changes in students' attitudes towards science, the available data present a nuanced picture. For example, interventions lasting as little as two weeks have been associated with favourable outcomes [58, 65], in contrast to cases of significantly longer interventions that failed to produce notable improvements in students' attitudes [1]. To reconcile these discrepancies, it would be prudent to assess the frequency of sessions delivered in terms of effective teaching hours, rather than simply considering the duration in weeks. However, the analysis of such data is hampered by the fact that many documents do not provide this information. Furthermore, the results of the present study show a consistent pattern among interventions that did not produce significant changes in students' attitudes: they were predominantly delivered in conventional classroom settings, lacked an evaluation process for the intervention design, and did not specify whether an evaluation of the implementation process was carried out in line with the study's objectives [1, 66–70] with only one exception.

Ultimately, the research questions served as a lens through which to identify key challenges and recommendations from researchers in their respective studies. While these data were not prevalent in the majority of documents, probably because the focus was on establishing the presence of an effect by presenting an independent variable rather than scrutinising the educational intervention itself, the pertinent elements that were identified are discussed below.

Researchers have identified a central challenge related to the difficult task of shaping students' attitudes towards science [67–69, 79]. Some scholars highlight the complexity of changing affective dimensions such as attitudes, a claim supported by cases where interventions led to increases in other variables such as academic performance rather than the targeted attitudes [66, 67]. It is therefore clear that the allocation of time for this purpose is of paramount importance, requiring a clear delineation of its use within the educational intervention. Conversely, a critical gap that has been identified relates to the lack of adequate training in the proposed educational approach, whether for the educators tasked with leading the intervention or for the students to familiarise themselves with it [69, 75]. This shortcoming has been implicated in the failure to achieve the objectives related to attitudes towards science.

Coupled with the findings from the segment on the challenges of educational interventions, there is an imperative to assess the lasting impact of attitudes long after programme implementation [18, 65, 70] to tailor interventions specifically to minority cohorts [64], and to subject proposed instructions to pre-implementation scrutiny [61]. These instructions will act as a catalyst for further investigation, strengthening the aims of the trial and establishing central benchmarks for reporting standards for educational interventions. This aligns seamlessly with the assertion regarding the need for comprehensive delineation of interventions [19]. Ultimately, these recommendations stand as pillars to guide the development and evaluation of educational methodologies. Notably, our study reveals that only seven documents acknowledged the use of evaluation tools to assess programme implementation against objectives or intervention protocols [26, 54, 55, 59, 68, 75, 77], with only four considering tools or methods to assess programme design [56, 61, 65, 79].

## Conclusions

This scoping review has outlined and described the impact of the educational interventions aimed at fostering positive attitudes towards science. The analysis of the corpus of documents revealed a discernible distinction in the concepts used to explore students' attitudes, reflecting different objects of interest, such as attitudes towards science versus attitudes towards a science

course. While these may represent distinct phenomena, there is a pervasive lack of clarity regarding their delineation, as evidenced by their interchangeable use within certain documents reviewed. There is therefore an urgent need for further research into their operational definition, facilitating the establishment of differentiating dimensions or a hierarchical organisation of the constructs.

In this review, we also identified salient features of educational interventions that demonstrated different pedagogical approaches to fostering attitudes towards science. However, none of these approaches showed clear effectiveness for their intended purpose. Consequently, there is an urgent need to assess the impact of other components within educational interventions, such as the competence of those overseeing implementation, evaluation of the developmental process, or the duration of effective instruction—factors that significantly influence the achievement of stated goals. This underlines the importance of comprehensive evaluation and refinement to optimise the effectiveness of interventions targeting attitudes towards science.

The identification of the main challenges and recommendations identified by the primary researchers in their studies also underlines the need to critically assess the timing of the development of interventions aimed at fostering positive attitudes towards science. It also underscores the need for a proactive approach by both teachers and students to the proposed programme. Finally, the recognition of intrinsic student variables influencing their attitudes, particularly the nuanced factors of racial/ethnic identity and self-efficacy, underscores the need for comprehensive exploration and elucidation. Remarkably, these variables have remained largely unaddressed in interventions, representing an untapped area for understanding the intricate interplay between these constructs and attitudes towards science. Thus, there is a compelling mandate to forge instructional designs that conscientiously incorporate and address these elements in order to realise their desired outcomes.

## Limitations and implications for research

While the research conducted has provided answers to the questions posed, it also reveals limitations in explaining the results obtained in terms of the effects of interventions. The lack of clarity in some studies about the constructs addressed, or the omission of potential contributing factors, directly undermines their validity and consequently the clarity of the findings presented. In addition, methodological limitations or the neglect of risks of bias within the included evidence were not considered. Although these considerations may not be in line with the prescribed literature review methodology, their importance cannot be overestimated, as they have a significant impact on the practical implications derived from the study.

Furthermore, a significant challenge lies in understanding the realities across different when implementing programs aimed at fostering positive attitudes toward science, as certain regions are notably underrepresented in the study´s results. For instance, in Africa and Europe, studies were identified in only three out of more than 50 countries each, an insufficient representation given the geographical size and population of these continents. In the Americas, data from key countries such as Mexico, Chile, or Argentina are conspicuously absent, despite these nations producing a substantial volume of scientific publications in education according to the Web of Science database during the time frame specified for this research. A closer examination of this database also revealed that Brazil, for example, is a significant contributor to scientific publications in the education category in the Americas. However it is plausible that exclusion criteria, such as language or database limitations, may have inadvertently excluded relevant sources that otherwise met the established search parameters, further restricting the study´s comprehensive scope. This gap limits our ability to fully understand the range of existing initiatives, their design specificities, and the

components included in educational interventions aimed at cultivating positive attitudes towards science.

While we are aware that this may have resulted in the exclusion of relevant studies due to language or other factors, adhering to the inclusion and exclusion criteria defined in the research protocol was essential for maintaining the coherence and validity of the findings.

Beyond the challenges posed by the constructs' clarity or geographic representation, further limitations arose from the chosen methodology. While the scoping review approach inherently allows for the inclusion of a wide range of literature that surpass purely systematic and rigorous evaluation processes [53], accessing sources beyond research articles proved challenging. As a result, the document sample in this study is relatively modest compared to other analogous studies [49, 51] or alternative types of literature reviews addressing the same topic [21]. This limitation may be primarily due to the lack of research objectives specifically aimed at analysing or characterising the components of educational interventions, a situation previously highlighted. In addition, it is plausible that the search strategies used in this study inadvertently limited the discovery of documents of different formats, such as handbooks or books.

The findings of this study have drawn attention to the key components that underpin educational interventions in science education, and serve as a precursor to a more in-depth investigation assessing their individual impact on intervention effectiveness. It is imperative to accentuate the paucity of literature addressing the design features within instructional programmes, emphasizing the need for further investigation, as echoed by other scholars [19, 92]. This endeavour holds promise for the delineation of essential indicators that are critical for the comprehensive reporting of the implementation of educational interventions.

In further detail, we advocate for the inclusion of follow-up evaluations after the completion of the intervention to measure the lasting impact on attitudes towards science. Remarkably, our findings revealed that such evaluations have only been conducted in two cases [63, 72]. We contend that these measures can strengthen an evolving research frontier, and subsequently promote a deeper understanding of cultivating positive attitudes towards science—an essential step towards promoting scientific literacy in our students.

## Supporting information

**S1 Protocol. Protocol of the systematic review.**
(DOCX)

**S1 Table. Summary data table.**
(XLSX)

**S1 Appendix. Data concentration.**
(DOCX)

## Author Contributions

**Conceptualization:** Noé Manuel García-Pérez, Gonzalo Peñaloza.

**Formal analysis:** Noé Manuel García-Pérez, Gonzalo Peñaloza.

**Investigation:** Noé Manuel García-Pérez, Gonzalo Peñaloza.

**Methodology:** Noé Manuel García-Pérez, Gonzalo Peñaloza.

**Writing – original draft:** Noé Manuel García-Pérez, Gonzalo Peñaloza.

**Writing – review & editing:** Noé Manuel García-Pérez, Gonzalo Peñaloza.

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
