## [Decision Letter · Decision Letter 0]

6 Sep 2024

PONE-D-24-19303A scoping review of interventions about middle school students' attitudes toward sciencePLOS ONE

Dear Dr. Peñaloza,

Thank you for submitting your manuscript to PLOS ONE. After careful consideration, we feel that it has merit but does not fully meet PLOS ONE’s publication criteria as it currently stands. Therefore, we invite you to submit a revised version of the manuscript that addresses the points raised during the review process.

We look forward to receiving your revised manuscript.

Kind regards,

Sonia Vasconcelos, PhD

Academic Editor

PLOS ONE

Journal Requirements:

2. Thank you for stating the following in your manuscript: 

"This work was supported by the National Council for Science and Technology (CONACYT) in

Mexico"

5. We note that Figure 3 in your submission contain map images which may be copyrighted. All PLOS content is published under the Creative Commons Attribution License (CC BY 4.0), which means that the manuscript, images, and Supporting Information files will be freely available online, and any third party is permitted to access, download, copy, distribute, and use these materials in any way, even commercially, with proper attribution. For these reasons, we cannot publish previously copyrighted maps or satellite images created using proprietary data, such as Google software (Google Maps, Street View, and Earth). For more information, see our copyright guidelines: http://journals.plos.org/plosone/s/licenses-and-copyright.

1) You may seek permission from the original copyright holder of Figure 3 to publish the content specifically under the CC BY 4.0 license.  

2) If you are unable to obtain permission from the original copyright holder to publish these figures under the CC BY 4.0 license or if the copyright holder’s requirements are incompatible with the CC BY 4.0 license, please either i) remove the figure or ii) supply a replacement figure that complies with the CC BY 4.0 license. Please check copyright information on all replacement figures and update the figure caption with source information. If applicable, please specify in the figure caption text when a figure is similar but not identical to the original image and is therefore for illustrative purposes only.

**Additional Editor Comments:**

When you address the comments of each reviewer, please pay particular attention to the necessary clarifications  regarding the accuracy of your analysis from the scoping review.

Reviewers' comments:

Reviewer's Responses to Questions

**Comments to the Author**

1. Is the manuscript technically sound, and do the data support the conclusions?

Reviewer #1: Yes

Reviewer #2: Yes

2. Has the statistical analysis been performed appropriately and rigorously? 

Reviewer #1: N/A

Reviewer #2: N/A

3. Have the authors made all data underlying the findings in their manuscript fully available?

Reviewer #1: Yes

Reviewer #2: Yes

4. Is the manuscript presented in an intelligible fashion and written in standard English?

Reviewer #1: Yes

Reviewer #2: Yes

5. Review Comments to the Author

Reviewer #1: The manuscript describes a scoping review aiming to characterise the research on student`s attitudes towards science involving the implementation of educational interventions. The work is well developed, the authors identified the constructs and the aspects addressed in educational interventions based in 37 published papers. Also, this research underlines the need to assess the impact of pedagogical interventions components, in order to clarify the effectiveness of these practices.

Nevertheless, the limitation of the work, related to the representative aspect of data, was poorly developed. The authors stated that 38% of studies were carried out in Turkey. This country, together with USA, account for 51,3% of the studies analysed. First of all, the authors should elaborate an explanation about this trend in Turkey. Is it related to a specific author and/or institution? In order to have a deeper understanding of the state of the art, the study would be expanded to capture, at least, the english abstracts from texts written in other languages, aiming to cover papers published in not-speaking Spanish/English countries. The authors pointed out, in the discussion sector (page 27), that a small number of studies were conducted in Latin America. But what about Brazil, a Portuguese-speaking country? It is the largest country in Latin America with a total area of over 8.5 million square kilometers. Its scientific production cannot be neglected.The authors presented a little discussion about the subject in the conclusion sector (page 31). They highlighted that Brazil is one of the countries with a significatif number of publications in the education area. I suggest to merge this topic in only one paragraph. Also, the authors must explicite that the results may be underestimated due to language limitation and this limitation should be highlighted in the abstract.

By the way, the abstract should be more precise and informative about the data found in the review. I list some of them:

- An important data pointed by the authors is that only 19% of the studies presented a conceptual definition of the construct of interest towards science (feelings, beliefs and values). This is a main issue in this work, which highlighted the opportunity to point out the necessity to adopt precise theoretical definitions in the study of attitudes.

- All studies used a questionnaire or Likert scale to collect data, being the majority of studies approached by a quantitative perspective.

- A medium of 4 articles had been published since 2014. Nevertheless, it increased to 8 in 2022. Does it reflect a necessity to study about trusting in science, an important issue by covid times?

- This review found 7 different teaching approaches to develop students` attitudes towards science, and 30,3% correspond to learning environments mediated by information and communication technologies, but the effectiveness of this teaching strategy remains inconclusive.

Other commentaries:

- It is necessary to characterise the ERIC database.

- Pg. 7 : k-12 – what does it mean? / Pg.9 : through

- The word “highlight/highlighted” is employed several times.

-The authors did not cite the ROSE Project (The Relevance of Science Education), based in the University of Oslo, Norway, and implemented in more than 40 countries. This Project is relevant in the research on students` attitude towards science.

Reviewer #2: The manuscript has certainly showcased extensive work on data collection and highlighted issues about educational interventions to promote positive attitudes towards science among middle school students, such as several gaps in the understanding of the constructs used in these interventions as well as the need to further research to focuses on intrinsic factors like racial background and perceptions on self-efficacy to better understand the long-term impact of these educational programs. Nonetheless, some issues in the presentation and discussion of the data should be addressed by the authors before publication so that the results presented in the manuscript can be better understood. Therefore, the following questions should be addressed by the authors:

1. In the Abstract (page 2, lines 6-8), the authors state “This paper presents a scoping review of 37 papers published in English and Spanish on the intervention literature from the last 10 years, both to describe the research outputs and to analyse the evidence on the effectiveness of different scientific databases”. I am not sure what the authors mean by “effectiveness of diferente scientific databases”. Please clarify.

2. Please revise the references format throughout the text. For example, in the introduction section (page 3, lines 11-12) reads: “Guo et al., 20-22; Musengimana et al., 20-21”. The hyphen should be removed between the year-numbers.

3. Still on page 3 (lines 16-18), the authors states: “In the words of Helvaci & Yilmaz (2022), the pedagogical practices developed by teachers have an impact on students' motivation and therefore promote the formation of positive attitudes towards science”. By the choice of wording by the authors, it is understood that the statement is a direct citation from Helvaci & Yilmaz. If so, quotation marks are needed.

4. In Table 1 the authors describe the pedagogical approaches used in science education. However, it is not clear within the table with theorical references were used to define each approach. I suggest revising the table to clarify this issue.

5. On page 6, the caption of Table 1 reads “Note. Original creation”. I suggest changing it to “Table created by the authors”. Same goes for the other tables in the manuscript.

6. On page 8 (lines 4-7), the authors state: “In the words of Savelsbergh et al. (2016), studying the constructs together may lead to confusing interpretations of the outcomes of an intervention, making it relevant to address them independently”. By the choice of wording by the authors, it is understood that the statement is a direct citation from Savelsbergh et al. If so, quotation marks are needed.

7. Still on page 8 (lines 10-13), reads: “Therefore, it may also be appropriate to analyse only those educational interventions that are implemented at the appropriate age for secondary education. For this reason, it may also be appropriate to analyse only those educational interventions that are implemented at the appropriate age for secondary education”. The phrase is duplicated. Please revise.

8. In the Method section (page 9), the authors describe the use of a “initial search equation” to explore concepts that would apply to the proposed scoping review. However, it is not clear throughout the manuscript nor in the supplementary information what these search equations were. Please clarify. The searching terms should be clearly described in the methodology section.

9. Regarding the figures and tables, all the figures and tables in the manuscript are presented without a caption. Figures should be followed by a descriptive caption describing the data shown and drawing attention to relevant findings or observations. Please revise.

10. Regarding Figure 3, I suggest revising the way the data is presented. The number of publications for each country should appear in the figure or be described in the caption (see comment above).

11. On page 18 (lines 6-7), reads: “The analysis shown in Figure 3 does not include documents that aim to conduct a literature review”. I believe that the referred analysis is the one shown in figure 4 and not figure 3 (as stated in the text). Please, revise.

12. Regarding Figure 4, I suggest including the number of each type of research design in the graph along with the percentages. Also, the caption is missing (see comment 9).

13. Figure 5, it is not clear what the category “ATPT” means, the caption is incomplete. It reads: “ATPT= Attitudes toward a particular”. Should it be “particular topic”? If so, what is the difference between this category and the previous one, ATSCD (particular science discipline)? Please clarify. Moreover, I suggest include the numbers related to each category in the graph and in the caption (see comment 9).

14. Figure 6, I suggest including the numbers on top of the bars in the graph to facilitate the reading of the data. Caption is also needed (see comment 9).

15. In the Discussion section, I encourage the authors to dig deeper into their interpretation of the data collected from their review, especially those regarding racial/ethnic identity and the lack of studies focusing on the topic. For example, the authors could explore the implications of those interventions that are under-researched, delving into why that might be the case and how it impacts practice or policies in science education. I strongly believe that exploring those points would benefit the discussion of the manuscript.

16. Also in the Discussion section (page 30, lines 19-22), reads: “Notably, our study reveals that only seven documents acknowledged the use of evaluation tools to assess programme implementation against objectives or intervention protocols, with only four considering tools or methods to assess programme design.” It should be stated which studies are the ones mentioned. Please revise.

17. In the Discussion section (pages 26-30) and later in the Conclusions (page 30-32) the authors highlight several limitations of their review. I suggest the authors to include a Limitation section after the conclusions so that they can explore in more details the implication of those limitations and how future studies could undermine such limitations.

18. In the supplementary 1 (attached to the manuscript), the authors describe their review protocol. In the protocol, one of the specific research questions is: “What is the current status of localised training programmes in terms of socio-demographic variables?’. However, in the description of the research questions in the manuscript (page 8), the above question is not mentioned. The question doesn’t appear to be addressed by the results presented in the manuscript either. Please, clarify.

19. Also in the supplementary 1, in the “Data extraction” section, it reads: “A summary data table is used to extract relevant information. The aim of this table is to systematically organise key findings in order to effectively address all research enquiries. The components to be included in the table are outlined below”. However, no table is presented. Please, provide table.

I would be happy to review a revised version of this manuscript.

6. PLOS authors have the option to publish the peer review history of their article (what does this mean?). If published, this will include your full peer review and any attached files.

Reviewer #1: **Yes: **Andréa Carla de Souza Góes

Reviewer #2: No

---

## [Author Response · Author response to Decision Letter 0]

24 Oct 2024

Response to reviewers

Dear reviewers:

Thank you for your valuable support and insightful recommendations regarding our manuscript. We have carefully addressed each suggestion with the aim of enhancing both the clarity and overall quality of our work. In the following paragraphs, we present your comments along with a detailed explanation or justification for the changes made, as well as the corrections we have implemented. Each reviewer´s feedback has been distinguished by a unique color, which is also reflected in the accompanying document, “Revised Manuscript with Track Changes”. Furthermore, in this document, we have included a superscript to indicate each modification made. Lastly, we have addressed the general comments provided by the journal´s academic editor, using a different color to distinguish them. 

We hope that the modifications made to our manuscript meet the journal criteria and that it will be considered for publication. In addressing the reviewers' and editor's comments, we have ensured that our responses were approached with both systematicity and rigor, maintaining the scientific integrity of our research. We are confident that the revised manuscript now reflects a thorough and well-supported analysis. Please let us know if any further adjustments are needed.

Reviewer 1: 

The manuscript describes a scoping review aiming to characterise the research on student`s attitudes towards science involving the implementation of educational interventions. The work is well developed, the authors identified the constructs and the aspects addressed in educational interventions based in 37 published papers. Also, this research underlines the need to assess the impact of pedagogical interventions components, in order to clarify the effectiveness of these practices.

1.Nevertheless, the limitation of the work, related to the representative aspect of data, was poorly developed. The authors stated that 38% of studies were carried out in Turkey. This country, together with USA, account for 51,3% of the studies analysed. First of all, the authors should elaborate an explanation about this trend in Turkey. Is it related to a specific author and/or institution? In order to have a deeper understanding of the state of the art, the study would be expanded to capture, at least, the english abstracts from texts written in other languages, aiming to cover papers published in not-speaking Spanish/English countries. 

Regarding this recommendation, after conducting a new search of relevant literature, we attempted to explain why the USA and Turkey accounted for more than half of the studies in our scoping review. In this regard, significant literature developed over the past decades on science education research showed these two countries established initiatives that allowed them to contribute more substantially to the field. We have reported this finding in our work. 

Furthermore, our findings were compared with the results of literature reviews identified in our study. We observed the same trend not only in the implementation of interventions aimed at developing attitudes toward science, but also in other key areas such as STEM education, assessment, or conceptual understanding. Special attention was given to the case of Turkey, where we found that science education research showed increasing interest since 1999, likely due to promotion and advancement policies and reforms in teaching training programs. This result was particularly surprising given that Turkey shares similar sociodemographic and economic characteristics with countries in our region that have not yet achieved similar results. However, support for education professionals and the increase in the number of journals publishing science education research have yielded positive results.

Concerning the suggestion to use English abstracts from texts written in other languages to cover papers published in not-speaking Spanish or English countries, we acknowledge that this approach could provide a broder representation of the science education field, particulalrly about intervention programs aimed at developing students attitudes. However, after analysing the corpus of documents in our study, we believe that this step would prevent us from fully exploring the research articles and adhering to our research objectives. It would be difficult to capture the full range of characteristics of the reported educational interventions as well as valuable recommendations or reflections of the main authors.

2. The authors pointed out, in the discussion sector (page 27), that a small number of studies were conducted in Latin America. But what about Brazil, a Portuguese-speaking country? It is the largest country in Latin America with a total area of over 8.5 million square kilometers. Its scientific production cannot be neglected. The authors presented a little discussion about the subject in the conclusion sector (page 31). They highlighted that Brazil is one of the countries with a significatif number of publications in the education area. I suggest to merge this topic in only one paragraph. Also, the authors must explicite that the results may be underestimated due to language limitation and this limitation should be highlighted in the abstract.

We acknowledge the significant contributions of countries where neither English nor Spanish is spoken. However, we also recognize the need to adhere to strict methodological guidelines to achieve the expected objectives. As a result, we included only articles written in English or Spanish as part of our inclusion criteria. This decision may have led to the exclusion of valuable information from other regions of the world, including Portuguese-speaking countries. To address this, we discussed the potential impact of this limitation in the discussion section, and, to underscore its importance, we created a specific limitations subsection in the conclusion. Additionally, we adopted the final recommendation and highlighted this issue in the abstract. 

3. By the way, the abstract should be more precise and informative about the data found in the review. I list some of them:

- An important data pointed by the authors is that only 19% of the studies presented a conceptual definition of the construct of interest towards science (feelings, beliefs and values). This is a main issue in this work, which highlighted the opportunity to point out the necessity to adopt precise theoretical definitions in the study of attitudes.

- All studies used a questionnaire or Likert scale to collect data, being the majority of studies approached by a quantitative perspective.

- A medium of 4 articles had been published since 2014. Nevertheless, it increased to 8 in 2022. Does it reflect a necessity to study about trusting in science, an important issue by covid times?

- This review found 7 different teaching approaches to develop students` attitudes towards science, and 30,3% correspond to learning environments mediated by information and communication technologies, but the effectiveness of this teaching strategy remains inconclusive.

We made several revisions to the abstract to address this recommendation as thoroughly as possible and to present some of the results in a more illustrative manner. However, due to the word limit stated for this section, it was not entirely feasible to include all the details. Nevertheless, we have strived to provide comprehensive coverage of the essential elements required in the abstract, ensuring that the key aspects of the study are clearly communicated.

4. It is necessary to characterise the ERIC database.

We adopted the suggested change and described how we initially used search terms to refine subsequent search equations. This phase was carried out using the ERIC and SciELO databases.

5. Pg. 7 : k-12 – what does it mean? 

We clarified in the text the term indicated. 

Pg.9 : through

6. The word “highlight/highlighted” is employed several times.

We adopted the suggested change and modified our text considering adequacy and coherence.

7. The authors did not cite the ROSE Project (The Relevance of Science Education), based in the University of Oslo, Norway, and implemented in more than 40 countries. This Project is relevant in the research on students` attitude towards science.

We recognize the relevance and impact of ROSE given that it let us know about several factors that have a bearing on students' attitudes toward science and technology and their motivation to learn in these areas. Nonetheless, it does not have an interventionist approach, which prevents it from being included in our study.

Reviewer 2

The manuscript has certainly showcased extensive work on data collection and highlighted issues about educational interventions to promote positive attitudes towards science among middle school students, such as several gaps in the understanding of the constructs used in these interventions as well as the need to further research to focuses on intrinsic factors like racial background and perceptions on self-efficacy to better understand the long-term impact of these educational programs. Nonetheless, some issues in the presentation and discussion of the data should be addressed by the authors before publication so that the results presented in the manuscript can be better understood. Therefore, the following questions should be addressed by the authors:

Dear reviewer. We greatly appreciate your support and the insightful recommendations you provided for our manuscript. We have carefully addressed each suggestion intending to enhance the clarity and quality of our work. For each recommendation, we have detailed the specific changes made and provided explanations to elucidate how these revisions improve the manuscript. We believe these modifications have significantly strengthened our submission. Thank you once again for your valuable feedback and for contributing to the improvement of our work.

1. In the Abstract (page 2, lines 6-8), the authors state “This paper presents a scoping review of 37 papers published in English and Spanish on the intervention literature from the last 10 years, both to describe the research outputs and to analyse the evidence on the effectiveness of different scientific databases”. I am not sure what the authors mean by “effectiveness of different scientific databases”. Please clarify.

We have revised the paragraph to correct a mistake. Originally, we mistakenly stated that the scoping review would be useful “to analyse the evidence on the effectiveness of different scientific databases”, which shifted the focus away from our intended objective. Instead, the review is designed to examine intervention programs. In addition to assessing the effectiveness of these programs, we aim to highlight the review's utility in identifying key characteristics in the design and implementation of such educational proposals. This focus aligns with the study´s main objective and research methodology.

2. Please revise the references format throughout the text. For example, in the introduction section (page 3, lines 11-12) reads: “Guo et al., 20-22; Musengimana et al., 20-21”. The hyphen should be removed between the year-numbers. 

Thank you for your advise. The reference format was checked throughout the entire text and corrected when needed. As a result of this process, we also corrected another reference at the end of the discussion section. 

3. Still on page 3 (lines 16-18), the authors states: “In the words of Helvaci & Yilmaz (2022), the pedagogical practices developed by teachers have an impact on students' motivation and therefore promote the formation of positive attitudes towards science”. By the choice of wording by the authors, it is understood that the statement is a direct citation from Helvaci & Yilmaz. If so, quotation marks are needed.

Upon reviewing the cited reference, we realized that a direct citation was not the most accurate way to represent the authors´ ideas. As the authors do not explicity state what we originally claimed, we have adjusted the text accordingly. We also tried to improve the connection between paragraphs.

4. In Table 1 the authors describe the pedagogical approaches used in science education. However, it is not clear within the table with theorical references were used to define each approach. I suggest revising the table to clarify this issue.

After revising the section, we made several changes to produce a more polished and appropriate version of Table 1, along with the preceding paragraph. We reorganize the ideas to better highlight the theoretical references used to define the teaching approaches presented in the table. Additionally, we justified the decision to combine the two sets of categories by emphasizing the need for flexibility in interpretation, particularly when adapting to the complexities of educational settings. This approach acknowledges that different models can be employed to suit the evolving nature of the field. Finally, we added a new column to the table to be able to reference each of the approaches within the text.

5. On page 6, the caption of Table 1 reads “Note. Original creation”. I suggest changing it to “Table created by the authors”. Same goes for the other tables in the manuscript. 

We adopted the suggested change.

6. On page 8 (lines 4-7), the authors state: “In the words of Savelsbergh et al. (2016), studying the constructs together may lead to confusing interpretations of the outcomes of an intervention, making it relevant to address them independently”. By the choice of wording by the authors, it is understood that the statement is a direct citation from Savelsbergh et al. If so, quotation marks are needed. 

The authors do not explicitly state what we initially claimed, so we have revised and modified the text accordingly. 

7. Still on page 8 (lines 10-13), reads: “Therefore, it may also be appropriate to analyse only those educational interventions that are implemented at the appropriate age for secondary education. For this reason, it may also be appropriate to analyse only those educational interventions that are implemented at the appropriate age for secondary education”. The phrase is duplicated. Please revise. 

 We corrected the identified error.

8. In the Method section (page 9), the authors describe the use of a “initial search equation” to explore concepts that would apply to the proposed scoping review. However, it is not clear throughout the manuscript nor in the supplementary information what these search equations were. Please clarify. The searching terms should be clearly described in the methodology section. 

Following this suggestion, we included the search equations used in the initial database searches (ERIC and SciELO). We ensured consistency with the description of how terms and specifications were added in subsequent database searches. Aditionally, we explained the rationale behind the selection of terms in the initial search. The scoping review methodology emphasizes the need to reflect the ‘Population’, ‘Concept’, and ‘Context’ in our search terms to establish a priori inclusion and exclusion criteria.

9. Regarding the figures and tables, all the figures and tables in the manuscript are presented without a caption. Figures should be followed by a descriptive caption describing the data shown and drawing attention to relevant findings or observations. Please revise.

We adopted the suggested change and added captions and legends when needed. 

10. Regarding Figure 3, I suggest revising the way the data is presented. The number of publications for each country should appear in the figure or be described in the caption (see comment above).

We adopted the suggested change.

11. On page 18 (lines 6-7), reads: “The analysis shown in Figure 3 does not include documents that aim to conduct a literature review”. I believe that the referred analysis is the one shown in figure 4 and not figure 3 (as stated in the text). Please, revise.

 We corrected the identified error. 

12. Regarding Figure 4, I suggest including the number of each type of research design in the graph along with the percentages. Also, the caption is missing (see comment 9).

We adopted the suggested change.

13. Figure 5, it is not clear what the category “ATPT

---

## [Decision Letter · Decision Letter 1]

2 Dec 2024

A scoping review of interventions about middle school students' attitudes toward science

PONE-D-24-19303R1

Dear Dr. Peñaloza,

We’re pleased to inform you that your manuscript has been judged scientifically suitable for publication and will be formally accepted for publication once it meets all outstanding technical requirements.

Kind regards,

Sonia Vasconcelos, PhD

Academic Editor

PLOS ONE

Reviewers' comments:

Reviewer's Responses to Questions

**Comments to the Author**

1. If the authors have adequately addressed your comments raised in a previous round of review and you feel that this manuscript is now acceptable for publication, you may indicate that here to bypass the “Comments to the Author” section, enter your conflict of interest statement in the “Confidential to Editor” section, and submit your "Accept" recommendation.

Reviewer #1: All comments have been addressed

Reviewer #2: All comments have been addressed

2. Is the manuscript technically sound, and do the data support the conclusions?

Reviewer #1: Yes

Reviewer #2: Yes

3. Has the statistical analysis been performed appropriately and rigorously? 

Reviewer #1: N/A

Reviewer #2: N/A

4. Have the authors made all data underlying the findings in their manuscript fully available?

Reviewer #1: Yes

Reviewer #2: Yes

5. Is the manuscript presented in an intelligible fashion and written in standard English?

Reviewer #1: Yes

Reviewer #2: Yes

6. Review Comments to the Author

Reviewer #1: The authors have fully addressed the reviewers comments.

Here are just 2 phrases to review:

... of science education interventions that (in) other regions (pg.28)

Furthermore, a significant challenge lies in understanding the realities across different (countries ?) when ... (pg.32)

Reviewer #2: In this reviewed version, the authors address the concerns previously raised. I commend the authors for their efforts in improving their manuscript and to address all my comments and suggestions. I consider the manuscript to be suitable for publication.

7. PLOS authors have the option to publish the peer review history of their article (what does this mean?). If published, this will include your full peer review and any attached files.

Reviewer #1: No

Reviewer #2: No

---

## [Editor Report · Acceptance letter]

23 Dec 2024

PONE-D-24-19303R1 

PLOS ONE

Dear Dr. Peñaloza, 

I'm pleased to inform you that your manuscript has been deemed suitable for publication in PLOS ONE. Congratulations! Your manuscript is now being handed over to our production team.

Kind regards, 

on behalf of

Dr. Sonia Vasconcelos 

Academic Editor

PLOS ONE